# Hydroxy-Selenomethionine Mitigated Chronic Heat Stress-Induced Porcine Splenic Damage via Activation of Nrf2/Keap1 Signal and Suppression of NFκb and STAT Signal

**DOI:** 10.3390/ijms24076461

**Published:** 2023-03-30

**Authors:** Yan Liu, Shenggang Yin, Ying He, Jiayong Tang, Junning Pu, Gang Jia, Guangmang Liu, Gang Tian, Xiaoling Chen, Jingyi Cai, Bo Kang, Lianqiang Che, Hua Zhao

**Affiliations:** 1Key Laboratory for Animal Disease-Resistance Nutrition, Ministry of Education/Institute of Animal Nutrition, Sichuan Agricultural University, 610000 Chengdu, China; 2College of Animal Science and Technology, Sichuan Agricultural University, Chengdu 611130, China

**Keywords:** chronic heat stress, spleen, antioxidant, inflammation, hydroxy-selenomethionine, pigs

## Abstract

Chronic heat stress (CHS) compromised the immunity and spleen immunological function of pigs, which may associate with antioxidant suppression and splenocyte apoptosis and splenic inflammation. Selenium (Se) exhibited antioxidant function and immunomodulatory through selenoprotein. Thus, this study aimed to investigate the protective effect of dietary hydroxy-selenomethionine (Selisso^®^, SeO) on chronic heat stress (CHS)-induced porcine splenic oxidative stress, apoptosis and inflammation. Growing pigs were raised in the thermoneutral environment (22 ± 2 °C) with the basal diet (BD), or raised in hyperthermal conditions (33 ± 2 °C) with BD supplied with 0.0, 0.2, 0.4 and 0.6 mg Se/kg SeO for 28 d, respectively. The results showed that dietary SeO supplementation recovered the spleen mass and enhanced the splenic antioxidant capacity of CHS growing pigs. Meanwhile, SeO activated the Nrf2/Keap1 signal, downregulated p38, caspase 3 and Bax, inhibited the activation of NFκb and STAT3, and enhanced the protein expression level of GPX1, GPX3, GPX4, SELENOS and SELENOF. In summary, SeO supplementation mitigates the CHS-induced splenic oxidative damages, apoptosis and inflammation in pigs, and the processes are associated with the activation of Nrf2/Keap1 signal and the suppression of NFκb, p38(MAPK) and STAT signal. It seems that the antioxidant-related selenoproteins (GPXs) and functional selenoproteins (SELENOS and SELENOF) play important roles in the alleviation processes.

## 1. Introduction

The livestock industry is facing a serious challenge due to climate change, where rising temperatures are causing heat stress in animals. This not only leads to decreased growth and increased disease susceptibility but also results in high mortality rates, contributing to billions of dollars in economic losses annually for the US livestock industry [1]. Pigs are extremely susceptible to hyperthermia due to limited evaporation and heat loss capacity resulting from lacking functional sweat glands. Prolonged heat stress which is called chronic heat stress (CHS) disrupt redox homeostasis and suppress immune system components [2], thereby increasing the risk of animal diseases, including an inflammatory reaction [3]. When animals are exposed to hyperthermia conditions, blood is directed from visceral tissues to peripheral tissues to maximize radiant heat dissipation [4], then fever and hypoxia-induced tissue injury and organ dysfunction occur [5,6]. The spleen is an important immune organ for the immune system, generating antibodies, regulating immune homeostasis, and serving as a reservoir for immune cells [7]. It has been reported that CHS cause atrophy of the spleen of a rat, which is attributed to the apoptosis and inflammation of the spleen [8].

Hydroxy selenomethionine (SeO) is a novel organic source of selenium with high efficiency and low toxicity [9,10]. Previous studies have demonstrated that dietary Se supplementation ameliorates the inflammatory response of tissue and organs under unfavorable conditions by regulating the antioxidant capacity and apoptosis, thereby improving the health of livestock [11,12]. The biological functions of selenium are achieved through selenoproteins, and currently, 25 selenoproteins have been identified in porcine species [11]. Most selenoproteins are involved in antioxidant and radical scavenging systems [13]. Hyperthermia-induced oxidative stress is one of the important factors leading to tissue and organ inflammation and apoptosis [14,15].

The nuclear factor erythroid 2-related factor 2 (Nrf2) is an emerging transcription factor that regulates cellular oxidative defense [16]. Kelch-like erythroid cell-derived protein with CNC homology-associated protein 1 (Keap1) serves as an Nrf2-binding protein, tethers Nrf2 in the cytoplasm and suppresses the activity of Nrf2 [17]. The ubiquitination-proteasomal degradation of Keap1 activates Nrf2 and promotes the expression of an array of antioxidant proteins, such as Heme oxygenase (HO-1), NAD(P)H: quinone oxidoreductase (NQO1) and Glutathione peroxidase (GPX) [16]. It has been confirmed that hyperthermia-induced oxidative stress can cause cell apoptosis and the response of apoptosis-related proteins such as p38, caspase 3, Bcl-2 and Bax [18,19]. Inflammation, oxidative stress and apoptosis are closely related, and several signaling pathways are involved in the stress and apoptosis-induced inflammatory response. Nuclear factor kappa B (NFκB) and signal transducer and activator of transcription 3 (STAT3) are important transcription factors that initiate the transcription of inflammatory mediator genes during inflammatory response [20,21].

Previous studies found that CHS causes cell apoptosis and functional abnormalities and the processes concomitant with a selenoprotein expression response [5,22,23]. Dietary Se supplementation mitigates LPS-induced murine splenic inflammatory response, and several abnormal expressed selenoprotein encoding genes return to normal [24]. Our previous study indicates that dietary Se supplementation mitigates CHS-induced depression of carcass traits and the meat quality of pigs [25]. The subsequent question was, therefore, to find whether those protective effects are partly attributed to the enhancement of the antioxidant capacity and immune function. Therefore, in the current study, the pig CHS model was further used to investigate: (1) the protective effect of dietary SeO supplementation on humoral immunity and splenic injury induced by CHS, and (2) the possible mechanistic link between the mitigation of CHS-induced splenic immune damage and the responses of apoptosis, Nrf2, NFκb pathway and selenoproteins.

## 2. Results

### 2.1. Serum Immunoglobulin and Inflammatory Level

As shown in Table 1, we investigated the effects of CHS and SeO on serum immune and inflammatory biomarkers. CHS tended to elevate serum immunoglobulin G (IgG) levels, and 0.6 mg Se/kg SeO supplementation significantly decreased serum IgG levels when compared with the CHS group (*p* < 0.05). Although CHS exhibited a limited effect on serum IgA, IgM, TNF-α and IL-6, 0.4 mg Se/kg SeO supplementation significantly decreased the serum IL-6 under CHS (*p* < 0.05).

### 2.2. Spleen Mass, Se Deposition, HSP70 mRNA and Protein Expression

As illustrated in Figure 1, the effect of CHS and SeO on spleen mass and Se deposition were investigated. The absolute and relative spleen weight were decreased under CHS (*p* < 0.001), and dietary SeO supplementation linearly recovered the spleen weight and index (*p* = 0.002) (Figure 1A,B). Although CHS alone did not affect the splenic Se deposition (*p* > 0.05), SeO supplementation linearly (*p* < 0.001) and quadratically (*p* = 0.005) facilitated Se deposition in the spleen (Figure 1C). As expected, CHS upregulated the protein abundance of HSP70 in the spleen, which was a major biomarker of heat stress. SeO supplementation linearly (*p* < 0.001) and quadratically (*p* = 0.003) decreased the HSP70 protein abundance in spleen tissue (Figure 1D,E). The *HSP70* mRNA expression response to CHS and dietary SeO exhibited a similar pattern to its protein; however, the difference among treatment groups was not statistically significant (Figure 1F).

### 2.3. Splenic Antioxidant Capacity

Direct assessment of splenic tissue antioxidant measurements suggested that CHS compromised the splenic antioxidant by decreasing T-AOC (*p* < 0.05) and numerically decreasing GSH-Px while increasing the MDA content (*p* < 0.05) (Figure 2). Dietary SeO supplementation linearly (*p* < 0.05) enhanced splenic GSH-Px and T-AOC activity, while linearly (*p* < 0.001) and quadratically (*p* = 0.019) decreased the splenic MDA content (Figure 2A,B,D). Apart from that, CHS and SeO supplementation did not statistically affect splenic T-SOD activity (Figure 2C).

### 2.4. The Protein Abundance Related to the NRF2 Signal

As shown in Figure 3, we further detected the effect of CHS and SeO on the protein expression level related to the NRF2 signal. CHS downregulated the protein expression abundance of HO-1 and NQO1 in the spleen (*p* < 0.05). All levels of SeO supplementation recovered or enhanced HO-1 protein expression abundance (*p* < 0.05), and 0.4 and 0.6 mg Se/kg SeO supplementation recovered NQO1 protein abundance to control level (*p* > 0.05) (Figure 3A,B). Although CHS did not affect the protein expression level of Nrf2 and Keap1, 0.2 and 0.4 mg Se/kg SeO supplementation upregulated the protein expression level of Nrf2 (*p* < 0.05) (Figure 3A,B); meanwhile, 0.4 and 0.6 mg Se/kg SeO decreased the Keap1 protein abundance (*p* < 0.05) (Figure 3A,B).

### 2.5. The Protein Expression of Apoptosis-Related Proteins

As showcased in Figure 4, the expressions of apoptosis-associated proteins (p38, Bax, Bcl-2 and caspase 3) were assessed in spleen tissue. CHS increased the protein abundance of caspase 3 and phosphorylation of p38 in spleen tissue (*p* < 0.05), which is a marker for apoptosis (Figure 4A,D). An amount of 0.4 and 0.6 mg Se/kg SeO supplementation down-regulated the protein abundance of casepase3 and proapoptotic protein Bax (*p* < 0.05) (Figure 4B,D). The anti-apoptosis protein Bcl-2 level was not changed under CHS and SeO addition in spleen tissue (*p* > 0.05) (Figure 4C).

### 2.6. The Phosphorylation Levels of NFκb and STAT Signal

As shown in Figure 5, to reveal the protective mechanisms of SeO supplementation on the splenic inflammation under CHS, the phosphorylation levels of NFκb and STAT signal-related proteins were detected. CHS upregulated the phosphorylation of NFκb, Iκb and STAT1 in spleen tissue (*p* < 0.05), and the phosphorylation of STAT3 was also upregulated numerically. SeO supplementation modestly or significantly downregulated the phosphorylation level of NFκb and STAT3, 0.4 and 0.6 mg Se/kg SeO decreased or recovered the phosphorylation levels of all these 4 signal proteins (*p* < 0.05).

### 2.7. The mRNA Expression of Inflammatory Response-Related Genes

A total of 10 inflammatory response-related gene expression profiles in the spleen were indicated in Figure 6. Dietary SeO supplementation partially reverse the effects of CHS on the expression of these genes. CHS significantly upregulated 4 pro- or anti-inflammatory regulators (*ICAM-1*, *IL-6*, *IL-8* and *iNOS-1*) (*p* < 0.05), and 0.4 and 0.6 mg Se/kg SeO supplementation recovered the expression of *ICAM-1*, *IL-6* and *IL-8* to normal level (*p* < 0.05) (Figure 6A). CHS tended to upregulate mRNA expression of *IL-10* and *TGF-β* and SeO supplementation tended to recover or downregulate *IL-10, TGF-β* and *TNF-α* (*p* < 0.1) (Figure 6B). No significant effect of CHS and SeO on *IL-1β*, *MCP-1* and *IFN-β* was detected (*p* > 0.05) (Figure 6C).

### 2.8. The mRNA Expression of Splenic Selenoproteins

The mRNA expression profiles of 25 selenoprotein encoding genes were illustrated in Figure 7. Except for 6 selenoprotein encoding genes that were too close to the background to be interpreted or reported, CHS increased the mRNA expression of 9 selenoprotein encoding genes (*GPX1*, *GPX3*, *GPX4*, *SELENOS*, *SEPHS2*, *TXNRD2*, *SELENOT*, *SELENOF* and *SELENOI*) (*p* < 0.05); among these 9 genes, SeO supplementation recovered (*p* < 0.05) or tended to recover (*p* < 0.1) 6 to normal level (*GPX1*, *GPX3*, *GPX4*, *SELENOS*, *SEPHS2* and *SELENOF*) (Figure 7A), and the remaining 3 selenoprotein encoding genes indicated lack of response to SeO under CHS (Figure 7B) (*p* > 0.05). Beyond that, the expression changes of 10 selenoprotein encoding genes did not reach statistical significance in response to CHS and SeO supplementation (Figure 7C) (*p* > 0.05).

### 2.9. The Proteins Expression of Key Response Splenic Selenoproteins

We further examine the response of splenic selenoproteins at the post-translational level to CHS and SeO, 5 key selenoproteins were detected (GPX1, GPX3, GPX4, SELENOS and SELENOF), which were shown in Figure 8. CHS tended to downregulate (*p* < 0.1) GPX1, GPX3, GPX4 and SELENOS, and downregulated (*p* < 0.1) SELENOF in the spleen (Figure 8A–F). SeO supplementation enhanced or recovered all these 5 selenoproteins under CHS when compare to the CHS group (*p* < 0.05), and GPX1, GPX3 and GPX4 exhibited a dose dependence upregulation in response to dietary SeO levels (Figure 8A–C,F).

## 3. Discussion

Prolonged hyperthermia causes immunosuppression and increased morbidity in animals [2]. The spleen is the major organ of cellular immunity and humoral immunity [7]. Currently, the pig CHS model was developed, and the results suggest a connection between that CHS-induced hepatic metabolic disorder and the depressed meat quality of pigs [5,25]. In the present study, we further explore the impact of the CHS on the immunity of pigs using spleen collected from the same pig CHS model and found that pigs placed in prolonged hyperthermia exhibited lower spleen weight and index (Figure 1A,B), and the HSP70 protein abundance of splenic tissue was upregulated (Figure 1D,E), which implicated varying degrees of splenic damages induced by CHS at tissue and molecular levels. Similar results are reported in poultry species in that long-term heat stress led to atrophy of immune organs (thymus and spleen), and a significant increase in plasma IgG and IgM [26,27]. Although no statistical difference was detected in the humoral immune indicator (IgG, IgA, IgM) and serum inflammatory cytokines (TNF-α, IL-6), serum IgG was numerically increased under CHS in the present study. Relevant studies have confirmed that Se can be used by almost all immune tissues and cells; with increasing dietary Se intake, humoral immune capacity is enhanced and the inflammatory response induced by heat stress is slowed down [15,28]. Dietary SeO supplementation linearly increased the splenic tissue Se deposition, and spleen weight, index and splenic HSP70 expression were recovered linearly (Figure 1A–D), and the serum IgG and IL-6 concentration were also linearly decreased. These results indicate that the porcine splenic tissue and functional damages induced by CHS were modestly alleviated by dietary SeO supplementation.

Heat stress accelerates reactive oxygen species (ROS) generation, which can cause oxidative stress in tissue and organ [29]. The antioxidative enzyme system (such as SOD and GSH-Px) is the first line of antioxidant defense, and CHS led to organ injury accompanying the change in the activity of these enzymes [5]. In this study, CHS numerically decreased the splenic GSH-Px and T-SOD, and significantly depressed the splenic T-AOC, then led to higher splenic MDA content (Figure 2), consistently to the results of heat stress poultry [27,30], the oxidative stress of spleen occurred under CHS. Dietary Se supplementations are effective in alleviating HS or LPS-induced organ oxidative stress in pig and mice models [5,31]. As expected in the current study, with SeO supplementation increasing, the activity of splenic GSH-Px and T-AOC was linearly increased, and splenic MDA content was linearly decreased under CHS (Figure 2A,B,D). These results indicate that SeO supplementation relieves the splenic oxidative damage of pigs suffering from CHS.

The Keap1/Nrf2 signaling pathway plays a pivotal role in the cellular antioxidant defense system against oxidative stress [16]. When Keap1 is deactivated and releases Nrf2, the resulting increase in Nrf2 levels triggers antioxidant defense mechanisms by regulating the synthesis of GPX, NQO1 and HO-1, which work together to scavenge ROS and protect against oxidative stress [32,33]. NQO1 is one of the major antioxidant enzymes, which catalyze the reduction of quinone and semiquinone to stable hydroquinone [34]. HO-1 participates in the degradation of prooxidant heme, resulting in the production of antioxidants [35]. Previous studies have reported that super-nutritional selenium status activates different genes in the Nrf2-antioxidant pathway of rodents, humans and dairy cow arterial endothelial cells [36,37,38]. Consistently, the present study also detected that dietary SeO beyond nutrient requirements (0.4 and 0.6 mg Se/kg) effectively activated the Nrf2 pathway and enhanced the HO-1 and NQO1 protein expression abundance. These results indicated that the activation of the Keap1/Nrf2 pathway may be the potential mechanism of the splenic antioxidant capacity enhanced by SeO supplementation.

Oxidative stress played a signaling role in trigging organ tissue apoptotic death [39,40]. Results from the rodents HS model indicate that the rat germ cells apoptosis related to HS-induced oxidative stress [41]. Furthermore, oxidative stress induced apoptosis is related to p38(MAPK) pathways [42]. Bax and Bcl-2 are two major regulatory proteins associated with apoptosis that participate in the pro-apoptotic and anti-apoptotic processes, respectively [43]. The activation of caspase 3 in cells and tissues is an important hallmark of apoptosis induced by various apoptotic signals [43]. In this study, CHS increased the splenic Caspase 3 protein expression level and phosphorylation of p38, while 0.4 and 0.6 mg Se/kg SeO supplementation significantly downregulated the protein level of Caspase 3 and Bax (Figure 4A,D). Although the response of Bcl-2 was not significant, these results and the atrophy of the spleen indicated CHS-induced apoptosis, which is consistent with the results from the poultry and porcine HS model [2,44,45]. Although not statistically significant, there was a trend towards increased BAX protein expression in the group treated with 0.2 mg Se/kg SeO. This could be due to complex interactions between biological effects and signal pathways such as cytokines and RNA, or alternative mechanisms of regulation involving selenium-rich metabolism. In short, SeO supplementation alleviated the atrophy and splenocyte apoptosis of the pig spleen under CHS.

Apoptosis and inflammation are closely related, and inflammation can cause apoptosis [46]. NF-κb/Iκb and STAT are the master regulators of inflammation and regulated the transcription of cytokines, chemokines and acute phase proteins [47,48,49]. It has been reported that heat stress and Se deficiency both activated the NF-κb/Iκb pathway, thus increasing the mRNA expression of pro-inflammatory cytokines (*IL-1β*, *IL-6*, *IL-8*, *IL-17* and *iNOS*) in the spleen of pigs and broilers [50]. A similar result was previously reported in rodents; heat stress increases the phosphorylation of NF-κb of the spleen [19]. Transcriptomic results also showed that heat stress cause the up regulation of STAT1 and STAT3 in the pikeperch liver [51]. Consistently, the current study also found that CHS increased the phosphorylation of NF-κb and STAT1 of the spleen. Meanwhile, the expression of 4 pro-inflammation genes (*ICAM-1*, *IL-6*, *IL-8* and *INOS-1*) were upregulated (Figure 5 and Figure 6). This suggests that CHS activated the inflammation-related pathway, which contributed to the inflammation in the porcine spleen. Dietary SeO supplementation (especially 0.4 and 0.6 mg Se/kg) significantly inhibited the inflammation-related signal (NF-κb, STAT1 and STAT3), and the mRNA expression of *ICAM-1*, *IL-6* and *IL-8* were downregulated under CHS. In summary, SeO mitigated the splenic inflammation induced by CHS through NF-κb, p38 and STAT pathways.

Growing evidence shows that dietary Se supplementation protected the internal organ of livestock and poultry under CHS [5,13,45]. Many of the protective effects of Se are thought to be mediated by selenoproteins [11]. In this study, CHS upregulated the mRNA expression of 9 selenoprotein encoding genes (*GPX1*, *GPX3*, *GPX4*, *SELENOS*, *SEPSH2*, *TXNRD2*, *SELENOT*, *SELENOF* and *SELENOI*) in the spleen (Figure 7A,B), which is similar to our previous studies in vivo and in vitro [5,22]. SeO supplementation (0.4 and 0.6 mg Se/kg) recovered the expression of 5 selenoprotein encoding genes (*GPX1*, *GPX3*, *GPX4*, *SELENOS* and *SEPSH2*) (Figure 7A) to the control level. Of these recovered genes, GPX1, GPX3 and GPX4 belong to the antioxidant enzyme family GPXs [11]. SELENOS is an endoplasmic reticulum (ER) resident protein which are associated with the mitochondrial pathway apoptosis and immunity response, and also have an antioxidant function [52]. SEPSH2 is the key regulatory protein in selenoprotein synthesis [11]. TXNRD2 protects cells from oxidant stress during embryogenesis [53], while SELENOT, SELENOF and SELENOI participate in redox metabolism regulation [54]. SELENOF has also been demonstrated to function as a gatekeeper of immunoglobulins in spleen tissue [55]. Interestingly, while CHS upregulated the mRNA expression of some selenoprotein encoding genes, the protein abundance of GPX1, GPX3, GPX4 and SELENOS showed a decrease in response to CHS treatment (Figure 8). This opposite response of transcript and protein levels might indicate negative feedback regulation to antioxidant selenoprotein synthesis, which is consistent with previous research [5,56]. Although some genes (*TXNRD2*, *SELENOT*, *SELENOF* and *SELENOI*) did not show a significant response to SeO supplementation under the CHS condition, their upregulation under CHS might reflect the transcriptional regulation of antioxidants. These results combined with previous reports indicate that dietary Se supplementation can protect internal organs from oxidative damage by regulating selenoprotein expression. Nevertheless, 10 splenic selenoprotein encoding genes (*DIO1*, *TXNRD1*, *SELENOH*, *SELENOK*, *SELENOM*, *SELENON*, *SELENOO*, *MSRB1*, *SELENOP* and *SELENOW*) (Figure 7C) exhibited insensitivity to CHS and SeO supplementation.

In summary, CHS led to the abnormal expression of selenoprotein genes and decreased antioxidant-related selenoproteins in the spleen, but SeO supplementation partially recovered this abnormality and enhanced antioxidant-related selenoprotein protein abundance in CHS-induced splenic damages.

## 4. Materials and Methods

### 4.1. Animal, Experiment Design and Management

The animal experiment procedures used were approved by the Institutional Animal Care and Use Committee of the Sichuan Agricultural University (Ethics Approval Code: SCAUAC201808-3).

The diets and physical characteristics of the experimental pigs were reported previously [5,25]. Briefly, a total of 30 crossbreeds castrated boars (Landrace × Yorkshire) × Duroc aged 14 weeks, with an average live weight of 49.64 ± 2.48 kg, were randomly allocated into 5 treatments (CON, CHS, 0.2SeO, 0.4SeO and 0.6SeO) with 6 replicates per treatment and 1 pig per replicate (*n* = 6). The CON group was fed a basal diet without Se supplementation and raised in a thermoneutral environment (22 ± 2 °C). The following four treatment groups were fed a basal diet supplemented with selenium in the form of hydroxy-selenomethionine (SeO, Selisso^®^ Adisseo France S.A.S., Paris, France) at dosage (mg Se/kg diet): 0.0, 0.2, 0.4 and 0.6, and these 4 groups of animals were subjected to CHS (33 ± 2 °C). The basal diet was formulated to meet the nutritional requirements of the National Research Council (2012) for growing pigs weighing 50−75 kg (Appendix A) and the selenium content in experiment feeds was presented in our previous study [57]. All pigs had free access to diet and water and were penned individually in an artificial climate chamber, which allowed for temperature setting and control.

### 4.2. Blood and Tissue Collection

At d 28 after an overnight fast, the blood was collected in anticoagulant-free tubes from the jugular vein, centrifuged at 2500× *g* for 10 min at 4 °C, and then the serum was separated immediately and refrigerated at −20 °C. Pigs were sedated by electrical stunning and slaughtered by manual exsanguination, spleens were rapidly removed and weighed, and the spleen index was calculated as the percentage of body weight. Spleen samples were immediately dissected and rapidly frozen in liquid nitrogen and stored at −80 °C for biochemical and molecular analyses.

### 4.3. Blood Inflammatory Factor and Immunoglobulin

Serum IgA, IgM and IgG antibodies were detected using an automatic biochemistry analyzer (Model 3100; Hitachi, Tokyo, Japan). Serum TNF-α and IL-6 were analyzed with the commercially available enzyme-linked immunosorbent assay (ELISA) kits (Meimian, Yancheng, China) according to the manufacturer’s instructions. Each measurement was performed in duplicate.

### 4.4. Selenium Concentration in Spleen Tissue

The total Se concentration in spleen tissue was performed as described in the National Food Safety Standard GB 5009.93–2010 for the determination of selenium in food, and calculated according to the protocol described in the previous study [25].

### 4.5. Splenic Antioxidant Capacity Analyses

Tissue homogenates and total protein content of the spleen were performed as previously described [58]. Glutathione peroxidase (GSH-Px), total superoxide dismutase (T-SOD), total antioxidant capability (T-AOC) and malondialdehyde (MDA) were measured using commercial kits (Nanjing Jiancheng Bioengineering Institute, Nanjing, China).

### 4.6. Real-Time PCR Analyses

Total mRNA isolation and reverse transcription were performed using a commercial reagent kit (Invitrogen, Carlsbad, CA, USA), and quantified the RNA concentration using a NanoDrop spectrophotometer (Thermo Scientific, Waltham, MA, USA). Additionally, we verified the RNA integrity by running the samples on an agarose gel and checked for the presence of distinct 28S and 18S ribosomal RNA bands. The subsequent quantitative real-time PCR (Q-RT-PCR) was carried out using a QuantStudio 6Flex Real-Time PCR system (Applied Biosystems, Foster City, CA, USA) as described in our previous study [23]. The primer sequences used for the assayed genes of 10 inflammatory response-related genes, 25 selenoprotein encoding genes, and 2 house-keeping genes (*β-ACTIN* and *GAPDH*) were referenced in our previous study and shown in (Appendix A) [57]. The geNorm algorithm was used to determine the reference genes (*β-ACTIN* and *GAPDH*). The primers were diluted according to the dilution table from manufacturer (Sangon, Shanghai, China).

### 4.7. Western Blot Analyses

The spleen tissue samples were lysed in RIPA lysis buffer (Beyotime, Shanghai, China), and the protein concentration was detected using the BCA kit (Nanjing Jiancheng Bioengineering Institute, Nanjing, China). The protein extracts were denatured, then electrophoresed on 10–12% SDS-PAGE gel and blotted onto the PVDF membrane (Millipore, Abingdon, UK). The membranes were blocked by 5% non-fat milk and immunoblotted with primary antibodies and corresponding secondary antibodies, the primary antibodies were diluted in the antibody diluent (Yamei, Shanghai, China), and the concentration and catalog information are compiled in Appendix A. The bands were visualized and quantified with ChemiDoc Touch (BioRad, Hercules, CA, USA), and an Image Lab™ software system (BioRad, Hercules, CA, USA) was used to analyze the densitometric of Western blot bands. The ratio of the target protein to β-actin protein represented the relative abundance of each target protein.

### 4.8. Statistical Analysis

The experiment applied a complete random design (CRD) and the treatment was a one-way structure. The data were statistically analyzed using the PROC MIXED procedure statistical package of SAS9.4 (SAS Institute 2003, Cary, NC, USA). Tukey’s multiple treatment comparisons processed with the LSMEAN statement of SAS9.4 and the pdmix800 macro were used for the letter grouping obtained [59]. The effects of different doses of SeO supplementation were analyzed using ANOVA with orthogonal polynomial contrasts. For the different statistical tests, significance was declared at *p* ≤ 0.05 and tendency was declared at 0.05 < *p* ≤ 0.10, unless otherwise stated.

## 5. Conclusions

In summary, as shown in Figure 9, our findings suggest that CHS may have complex effects on the porcine spleen involving abnormal humoral immunity depression, oxidative stress and inflammation. Our data indicate that dietary Se supplementation in the form of SeO can activate the Nrf2/Keap1 pathway and enhance splenic antioxidant capacity. Additionally, our results suggest that increased dietary Se supplementation beyond nutritional requirements (0.4 and 0.6 mg Se/kg) may alleviate splenocyte apoptosis and inflammation through inactivating p38 MAPK, NFκB and STAT pathways in the spleen of pigs under CHS. Notably, our study reveals important protective roles of antioxidant-related selenoproteins (e.g., GPXs) and functional selenoproteins (e.g., SELENOS and SELENOF) in the splenic immune processes under CHS.

## Figures and Tables

**Figure 1 ijms-24-06461-f001:**
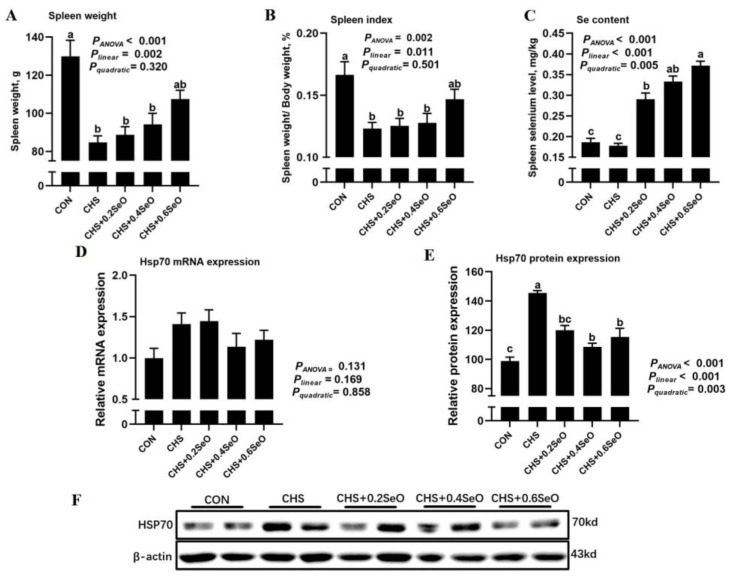
Effects of CHS and SeO supplementation on spleen weight (**A**), spleen weight/body weight (**B**), spleen Se concentration (**C**), mRNA (**D**) and protein (**E**,**F**) expression of HSP70 in the spleen of growing pigs. The results were expressed as mean ± SEM (*n* = 6). Different letters (a, b and c) denote significant differences, *P*_ANOVA_: *p* value of Analysis of Variance, *P*_linear_ and *P*_quadratic_: *p* value of Orthogonal Polynomial Contrasts.

**Figure 2 ijms-24-06461-f002:**
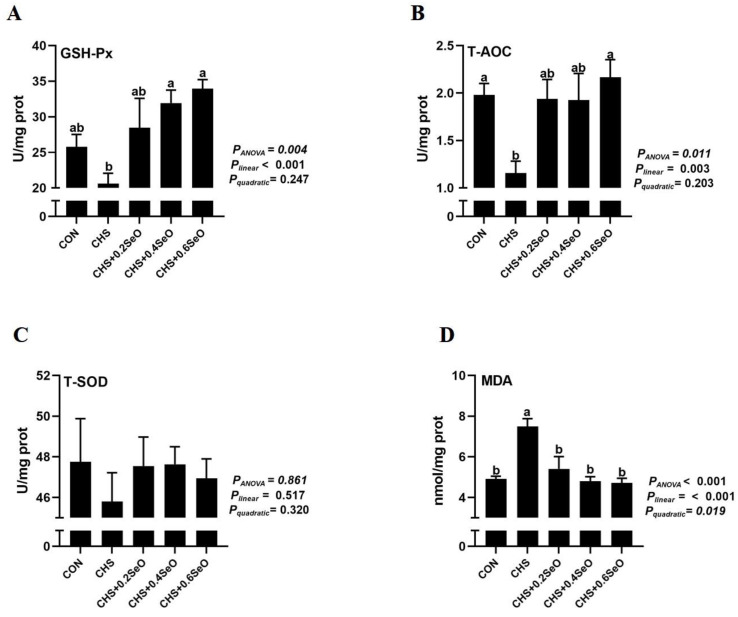
Effects of CHS and SeO supplementation on splenic antioxidant capacity of growing pigs. (**A**) GSH-Px, glutathione peroxidase; (**B**) T-AOC, total antioxidant capacity; (**C**) T-SOD, total superoxide dismutase; (**D**) MDA, malondialdehyde. The results were expressed as mean ± SEM (*n* = 6). Different letters (a and b) denote significant differences, *P*_ANOVA_: *p* value of Analysis of Variance, *P*_linear_ and *P*_quadratic_: *p* value of Orthogonal Polynomial Contrasts.

**Figure 3 ijms-24-06461-f003:**
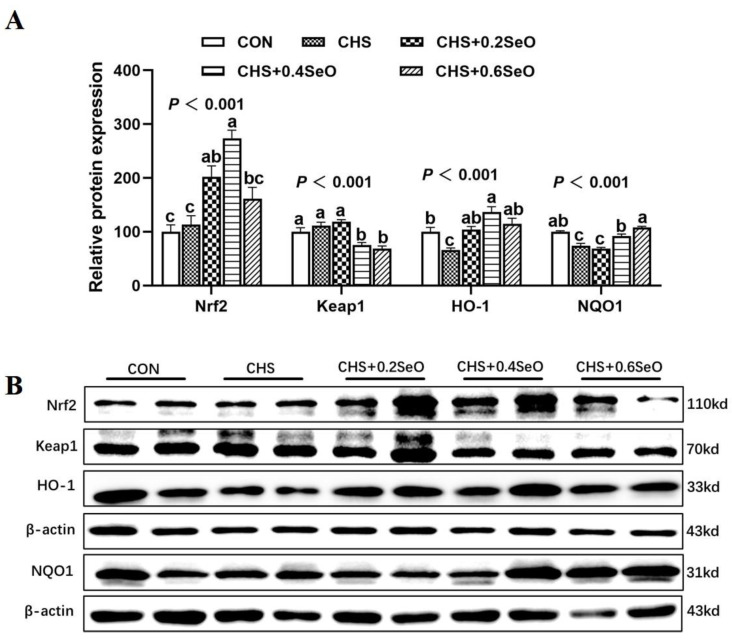
Effects of CHS and SeO supplementation on the abundance of protein related to Nrf2 signal in the spleen. (**A**) Relative protein expression level, (**B**) Relevant protein bands. The results were expressed as mean ± SEM (*n* = 6). Different letters (a, b and c) denote significant differences (*p* < 0.05).

**Figure 4 ijms-24-06461-f004:**
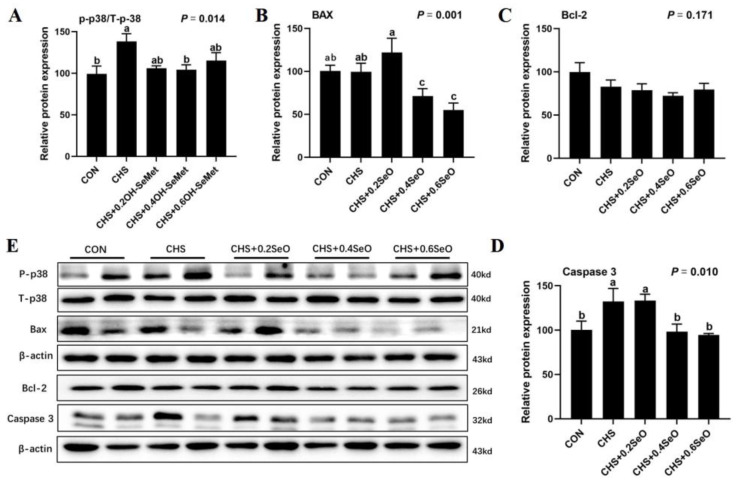
Effects of CHS and SeO supplementation on the abundance of protein related to apoptosis in the spleen. (**A**–**D**) Relative protein expression level, (**E**) Relevant protein bands. The results were expressed as mean ± SEM (*n* = 6). Different letters (a, b and c) denote significant differences (*p* < 0.05).

**Figure 5 ijms-24-06461-f005:**
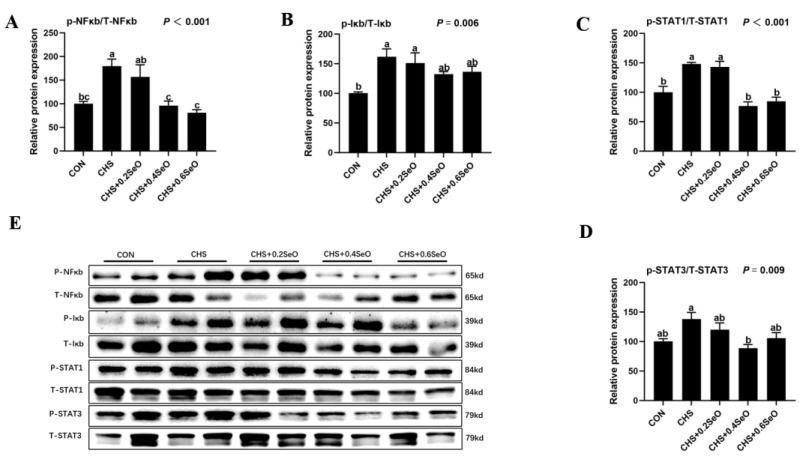
Effects of CHS and SeO supplementation on the abundance of protein related to NFκb and STAT signal in the spleen. (**A**–**D**) Relative protein expression level, (**E**) Relevant protein bands. The results were expressed as mean ± SEM (*n* = 6). Different letters (a, b and c) denote significant differences (*p* < 0.05).

**Figure 6 ijms-24-06461-f006:**
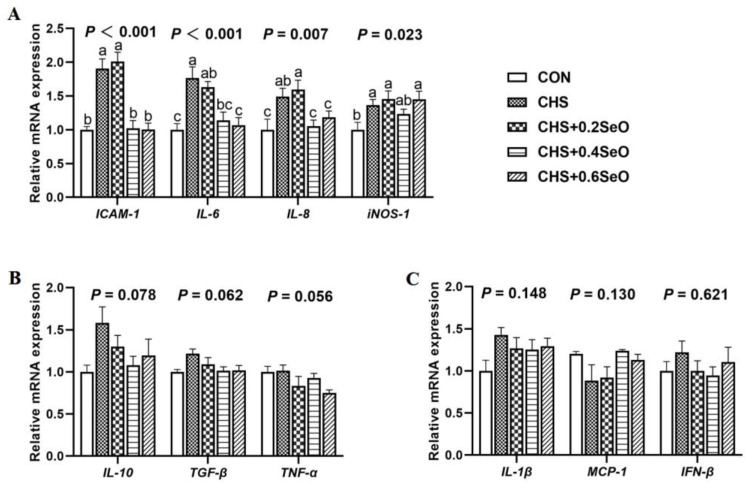
Effects of CHS and SeO supplementation on the mRNA expression of inflammation-related genes in the spleen. (**A**) Inflammation-related genes with significant differences, (**B**) Genes with trend-level differences, (**C**) Genes with no significant differences. The results were expressed as mean ± SEM (*n* = 6). Different letters (a, b and c) denote significant differences (*p* < 0.05).

**Figure 7 ijms-24-06461-f007:**
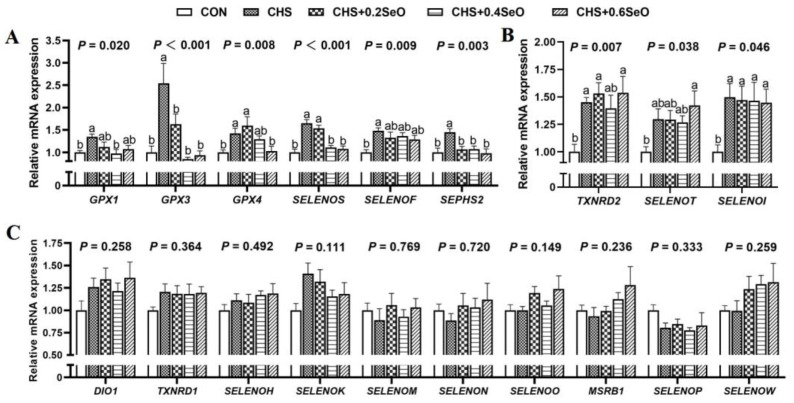
Effects of CHS and SeO supplementation on the mRNA expression of selenoproteins in the spleen. (**A**,**B**) selenoprotein genes with significant changes caused by CHS and SeO, (**C**) Genes with no significant difference. The results were expressed as mean ± SEM (*n* = 6). Different letters (a and b) denote significant differences (*p* < 0.05).

**Figure 8 ijms-24-06461-f008:**
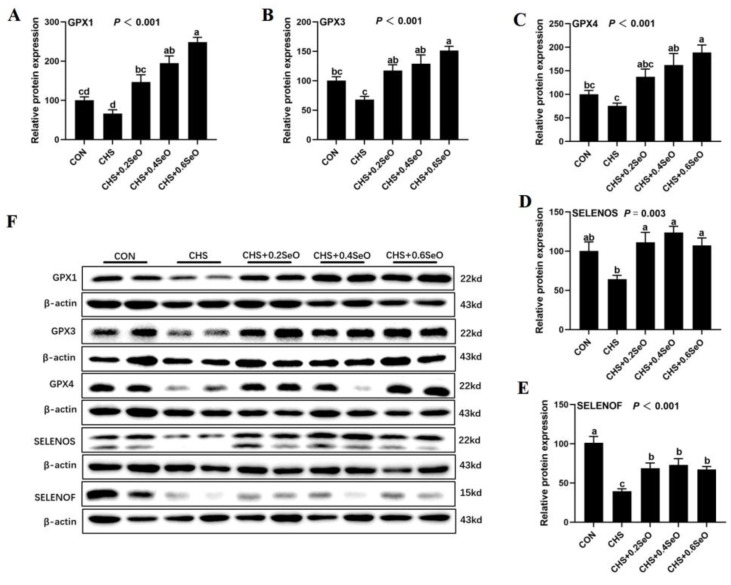
Effects of CHS and SeO supplementation on the protein expression abundance of GPX1, GPX3, GPX4, SELENOS and SELENOF in the spleen. (**A**–**E**) Relative protein expression level, (**F**) Relevant protein bands.The results were expressed as mean ± SEM (*n* = 6). Different letters (a, b and c) denote significant differences (*p* < 0.05).

**Figure 9 ijms-24-06461-f009:**
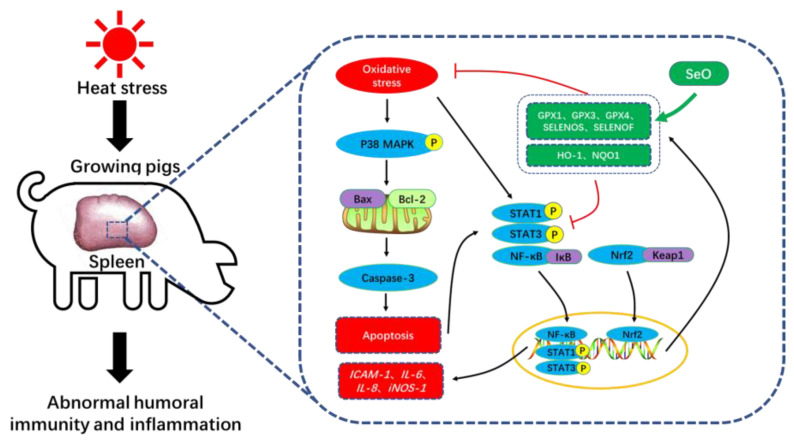
The potential mechanism of SeO mitigated CHS-induced spleen damage in growing pigs.

**Table 1 ijms-24-06461-t001:** Effects of CHS and SeO supplementation on serum immunoglobulin and inflammatory factors of growing pigs.

Item	Con	High-Temperature Conditions				SEM *	ANOVA *p*-Value	Linear	Quadratic
		CHS	0.2SeO	0.4SeO	0.6SeO				
IgA (mg/L)	49.54	46.67	46.66	42.23	43.16	0.97	0.103	0.133	0.827
IgG (g/L)	3.88 ^ab^	4.38^a^	4.32 ^a^	3.57 ^ab^	3.03 ^b^	0.14	0.002	<0.001	0.296
IgM (mg/L)	0.40	0.41	0. 43	0.41	0.45	0.01	0.432	0.198	0.488
TNF-α (ng/L)	0.41	0.43	0.36	0.45	0.45	0.01	0.115	0.233	0.228
IL-6 (ng/L)	1.10 ^a^	0.94 ^ab^	1.08 ^a^	0.82 ^b^	0.97 ^ab^	0.27	0.009	0.963	0.358

Notes. ^a b^ Mean values with different letters were significantly different *p* < 0.05. * SEM, total standard error of means (*n* = 6).

## Data Availability

Not applicable.

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
