# Peer review of "Hydroxy-Selenomethionine Mitigated Chronic Heat Stress-Induced Porcine Splenic Damage via Activation of Nrf2/Keap1 Signal and Suppression of NFκb and STAT Signal"

_ijms, 2023, doi:10.3390/ijms24076461_

Round 1
Reviewer 1 Report
Dear authors,
Thank you for submitting your interesting results. Overall, I found your writing to be well-structured and engaging. However, I believe that your work could benefit from including more up-to-date bibliographic sources. Strong theory foundations for any research or analysis are necessary, and utilizing more recent sources can help to ensure the relevance and accuracy of your work.
In addition, there were a few inaccuracies that I would like to point out. Specifically:
Line 22: "CON" is not the acronym of the basal diet. Please specify the acronym
Lines 25-27: Check the English
Lines 40-48: A reference linking prolonged heat stress to spleen dysfunction is missing here and it should be provided.
Table 1: why IgA was so high in the control group? And why IgA decreased in the CHS group? Authors should justify these aspects.
Figure 1: the letters should be also used in figure 1D to indicate probability if letters have been used in figure 1E. Letters "a", "b" and "c" should be also placed near the p-value, or their meaning should be provided in the caption.
Line 135: In general, the term "protein expression" should not be used. Are they protein levels or gene expression levels analyses? The term "protein expression" does not well describe the type of analysis, please check it in the entire paper. Moreover, the use of the verb "facilitate" is unappropriated, please change the verb.
Line 147: correct "caspase 3"
All paper's figures: please report the meaning of the letters used to indicate the probability.
Lines 167-172: CHS did not increase so much iNOS-1, and Selenium administration did not significantly decrease iNOS-1 gene expression compared to the CHS group as reported by the authors. Please better comment on this result.
Paragraph 2.7: authors should include the observation that the effects of CHS and selenium administration were different on the considered expression of analyzed genes.
Paragraph 2.8: Authors should scientifically motivate why they analyzed 25 selenoproteins in the spleen.
Paragraph 2.9: why authors did not analyze SEPHS2 protein levels? The gene expression analyses reported that CHS increased SEPHS2 expression while the selenium-treated animals displayed a significant reduction of SEPHS2 gene expression compared to the CHS group. A similar problem is present also for other genes. Authors can choose to provide the related protein level analyses or at least justify the absence of protein analyses.
Lines 244-246: this sentence is poorly written.
References: some references are quite old. Authors should provide a more recent bibliography (at least later than 2016)
Lines 259-264: commenting figure 4, authors should justify or comment why the administration of 0.2 SeO increased Bax protein levels in figure 4B
Line 291: authors should list the genes that recovered the expression after Se administration. In particular, authors showed that CHS upregulated some selenoproteins-related genes and downregulated others. Authors should better comment on this result, and try to motivate this result with literature.
Lines 302-305:this comment should be placed near the comments of related figures (figure 7, lines 288-292).
Line 302: "four selenoprotein encoding genes (TXNRD2, SELENOT, SELENOF, and SELENOI) were non-significant under the CHS condition", looking figure 7 is emerging an opposite result: these genes significantly increase their expression under CHS conditions.
Line 312: authors should write the 10 genes.
Materials and methods: authors should provide primer and antibodies companies and dilutions.
I would recommend you revise also the grammar and provide more comments to increase the relevance of your results.
I look forward to seeing the revisions to your work.
Best regards
Author Response
Dear Reviewer:
Thank you for your comments concerning our manuscript
ID: ijms-2282451 entitled “Hydroxy-selenomethionine mitigated chronic heat stress-induced porcine splenic damage via activation of Nrf2/Keap1 signal and suppression of NFκb and STAT signal”. Those comments are all valuable and very helpful for revising and improving our paper, as well as the important guiding significance to our researches. We have studied comments carefully and have made correction according to the comments as following:
Point 1: Thank you for submitting your interesting results. Overall, I found your writing to be well-structured and engaging. However, I believe that your work could benefit from including more up-to-date bibliographic sources. Strong theory foundations for any research or analysis are necessary, and utilizing more recent sources can help to ensure the relevance and accuracy of your work.
Response: We thank the reviewer’s good comments for this manuscript. We replaced part of overly outdated evidence with new bibliography which will not influence the content and framework of the paper. Related changes can be read in the newly uploaded manuscript.
Point 2: In addition, there were a few inaccuracies that I would like to point out. Specifically:
Line 22: "CON" is not the acronym of the basal diet. Please specify the acronym
Lines 25-27: Check the English
Response: Thanks to point out these issues. We have specified the acronym of basal diet and corrected the English issue in this paragraph.
Point 3: Lines 40-48: A reference linking prolonged heat stress to spleen dysfunction is missing here and it should be provided.
Response: Thanks for point out this issue, the relevant reference has been provided and cited in this sentence as follows:
- Huo, C.; Xiao, C.; She, R.; Liu, T.; Tian, J.; Dong, H.; Tian, H.; Hu, Y., Chronic heat stress negatively affects the immune functions of both spleens and intestinal mucosal system in pigs through the inhibition of apoptosis. Microbial pathogenesis 2019, 136, 103672.
Point 4: Table 1: why IgA was so high in the control group? And why IgA decreased in the CHS group? Authors should justify these aspects.
Response: We thank the reviewer’s good question; IgA is an immunoglobulin that plays a crucial role in the immune system's defense against infections. Prolonged or severe heat stress can have negative effects on the immune system, including a decrease in IgA levels in lambs . However, the inter-group difference in serum IgA content did not reach statistical significance in this study, so its significance was not further discussed.
Shi, L., Xu, Y., Mao, C. et al. Effects of heat stress on antioxidant status and immune function and expression of related genes in lambs. Int J Biometeorol 64, 2093–2104 (2020).
Point 5: Figure 1: the letters should be also used in figure 1D to indicate probability if letters have been used in figure 1E. Letters "a", "b" and "c" should be also placed near the p-value, or their meaning should be provided in the caption.
Response: Thanks for the reviewer’s advice. In this study we applied one-way ANOVA to test the differences between the treatment groups. When the results of one-way ANOVA were significant (P<0.05), the Turkey’s multiple comparisons was further processed and the significance between multiple comparisons was marked with different letters, otherwise the multiple comparisons were not meaningful. In Figure 1D the P value of ANOVA was 0.131, so the Turkey’s multiple comparisons was not processed.
Point 6: Line 135: In general, the term "protein expression" should not be used. Are they protein levels or gene expression levels analyses? The term "protein expression" does not well describe the type of analysis, please check it in the entire paper. Moreover, the use of the verb "facilitate" is unappropriated, please change the verb.
Line 147: correct "caspase 3"
Response: We thank the reviewer for the great help to point out above mistake in the manuscript, the “protein expression” have been changed to “abundance of protein” or “protein expression abundance”, and “facilitate” has been changed to “increased”, and “Caspase3” has been change to “caspase 3”. we have checked the full text and revised the similar issues.
Point 7: All paper's figures: please report the meaning of the letters used to indicate the probability.
Response: We thanks for the reviewer’s kindly check; we have added the information of letters in the lagend of all figures.
Point 8: Lines 167-172: CHS did not increase so much iNOS-1, and Selenium administration did not significantly decrease iNOS-1 gene expression compared to the CHS group as reported by the authors. Please better comment on this result.
Response: We thank the reviewer for point out this issue, Inducible nitric oxide synthase 1 (iNOS-1) is an enzyme that catalyzes the production of nitric oxide (NO) from L-arginine. It is an important regulator of immune response and inflammation in the body. In our study, Although the upregulation amplitude of iNOS-1 gene expression caused by CHS was not very high, statistical analysis showed that it was significantly higher than that in the control group (P = 0.023), and the statistical power exceeded 80%. The addition of SeO did not have a significant improvement effect on the upregulation of this gene, so it was not supported as the main conclusion.
Point 9: Paragraph 2.7: authors should include the observation that the effects of CHS and selenium administration were different on the considered expression of analyzed genes.
Response: Thank you for your valuable suggestion. We have revised the manuscript to include a statement in the Results section noting that CHS and selenium administration had varying effects on the expression of the analyzed genes. Thank you again for your guidance.
Point 10: Paragraph 2.8: Authors should scientifically motivate why they analyzed 25 selenoproteins in the spleen.
Response: Thank you for your valuable suggestion. We have further explained in the Introduction section why we chose to analyze 25 selenoprotein genes in the spleen. In brief, these proteins are essential antioxidants and immune modulators that are associated with numerous diseases and health conditions. Therefore, we believe that studying these genes is of utmost importance. Thank you again for your guidance.
Point 12: Paragraph 2.9: why authors did not analyze SEPHS2 protein levels? The gene expression analyses reported that CHS increased SEPHS2 expression while the selenium-treated animals displayed a significant reduction of SEPHS2 gene expression compared to the CHS group. A similar problem is present also for other genes. Authors can choose to provide the related protein level analyses or at least justify the absence of protein analyses.
Response: Thank you for your valuable suggestion. We agree with your comment that, based on the differences in SEPHS2 gene expression between CHS and selenium-treated animals, it would be beneficial to verify protein levels. We screened dozens of different suppliers' antibodies, including SEPHS2, but only antibodies against GPX1, GPX3, GPX4, SELENOS, and SELENOF were effective in our study. The specific functions of these selenoprotein genes will be further studied in future research.
Point 13: Lines 244-246: this sentence is poorly written.
Response: Thank you for your valuable suggestion. We have revised the sentence to “When Keap1 is deactivated and releases Nrf2, the resulting increase in Nrf2 levels triggers antioxidant defense mechanisms by regulating the synthesis of GPX, NQO1, and HO-1, which work together to scavenge ROS and protect against oxidative stress.”. In the revised manuscript, we further carefully reviewed all text to ensure accuracy and ease of understanding. Thank you again for your guidance.
Point 14: References: some references are quite old. Authors should provide a more recent bibliography (at least later than 2016)
Response: Thank you for your valuable suggestion. We have reviewed the references and update or removed some outdated literature in the revised manuscript to reflect recent research progress. Thank you again for your guidance.
Point 15: Lines 259-264: commenting figure 4, authors should justify or comment why the administration of 0.2 SeO increased Bax protein levels in figure 4B
Response: Thank you for your valuable suggestion. We agree that it may be necessary to provide an explanation in the manuscript about why administration of 0.2 SeO increased Bax protein levels in Figure 4B. In the revised manuscript, we have included the following explanation: Although not statistically significant, there was a trend towards increased BAX protein expression in the group treated with 0.2 mg Se/kg OH-SeMet. This could be due to complex interactions between biological effects and signal pathways like cytokines and RNA, or alternative mechanisms of regulation involving selenium-rich metabolism. Thank you again for your guidance.
Point 16: Line 291: authors should list the genes that recovered the expression after Se administration. In particular, authors showed that CHS upregulated some selenoproteins-related genes and downregulated others. Authors should better comment on this result, and try to motivate this result with literature.
Line 302: "four selenoprotein encoding genes (TXNRD2, SELENOT, SELENOF, and SELENOI) were non-significant under the CHS condition", looking figure 7 is emerging an opposite result: these genes significantly increase their expression under CHS conditions.
Lines 302-305: this comment should be placed near the comments of related figures (figure 7, lines 288-292).
Line 312: authors should write the 10 genes.
Response: Thank you for your valuable suggestion. We strongly agree with the comments provided by the reviewer. This paragraph has been extensively revised to address the comments raised by the reviewer. Such as:
“Growing evidence shows that dietary Se supplementation protected the internal organ of livestock and poultry under CHS [5, 15, 46]. Many of the protective effects of Se are thought to be mediated by selenoproteins [14]. In this study, CHS up-regulated the mRNA expression of 9 selenoprotein encoding genes (GPX1, GPX3, GPX4, SELENOS, SEPSH2, TXNRD2, SELENOT, SELENOF, and SELENOI) in the spleen (Figure 7 A and B), which is similar to our previous studies in vivo and vitro [5, 23]. OH-SeMet supplementation (0.4 and 0.6 mg Se/kg) recovered the expression of 5 selenoprotein encoding genes (GPX1, GPX3, GPX4, SELENOS, and SEPSH2) (Figure 7A) to the control level. Of these recovered genes, GPX1, GPX3 and GPX4 belong to the antioxidant enzyme family GPXs [14]. SELENOS is an endoplasmic reticulum (ER)-resident proteins which are associated with the mitochondrial pathway apoptosis and immunity response, and also have an antioxidant function [54]. SEPSH2 is the key regulatory protein in selenoprotein synthesis [14]. TXNRD2 protects cells from oxidant stress during embryogenesis [55], while SELENOT, SELENOF, and SELENOI participate in redox metabolism regulation [56]. SELENOF has also been demonstrated to function as a gatekeeper of immunoglobulins in spleen tissue [57]. Interestingly, while CHS upregulated the mRNA expression of some selenoprotein encoding genes, the protein abundance of GPX1, GPX3, GPX4, and SELENOS showed a decrease in response to CHS treatment (Figure 8), This opposite response of transcript and protein levels might indicate negative feedback regulation to antioxidant selenoprotein synthesis, which is consistent with previous research [5]. Although some genes (TXNRD2, SELENOT, SELENOF and SELENOI) did not show a significant response to OH-SeMet supplementation under the CHS condition, their up-regulation under CHS might reflect the transcriptional regulation of antioxidants. These results combined with previous reports indicate that dietary Se supplementation can protect internal organs from oxidative damage by regulating selenoprotein expression. Nevertheless, 10 splenic selenoprotein encoding genes (DIO1, TXNRD1, SELENOH, SELENOK, SELENOM, SELENON, SELENOO, MSRB1, SELENOP and SELENOW) (Figure 7 C) exhibited insensitivity to CHS and OH-SeMet supplementation.
In summary, CHS led to abnormal expression of selenoprotein genes and de-creased antioxidant-related selenoproteins in the spleen, but OH-SeMet supplementation partially recovered this abnormality and enhanced antioxidant-related selenoprotein protein abundance in CHS-induced splenic damages.”
Point 17: Materials and methods: authors should provide primer and antibodies companies and dilutions.
Response: Thank you for your valuable suggestion. In the revised manuscript, we added the primer and dilutions information to the Materials and Methods section, and the primer information has been listing in the Supplementary Table S3. We apologize for any inconvenience caused by our initial oversight and thank you for bringing this to our attention.
We tried our best to improve the manuscript and some mistakes also have been corrected. These changes will not influence the content and framework of the paper. And here we did not list all the changes in revised paper. We appreciate for Reviewer’s warm work earnestly, and hope that the correction will meet with approval.
Once again, thank you very much for your comments and suggestions.
Best wishes
Dr. Hua Zhao Corresponding Author

Reviewer 2 Report
I reviewed the manuscript entitled “Hydroxy-selenomethionine mitigated chronic heat stress-induced porcine splenic damage via activation of Nrf2/Keap1 signal and suppression of NFκb and STAT signal”.
The introduction is described sufficiently and introduces the reader to the topic of research.
The results are presented logically and allow the reader to follow the analysis course. I have a couple suggestions and questions:
-Line 87: Table 1 instead of Table 2
-Line 110: You are asked to give some more explanations about the differences between mRNA and protein levels of Hsp70 in 0.2 SeO group
-Figure 2: Please add the full name of GSH-Px, T-AOC, T-SOD, MDA in the figure caption
-Figure 3, 4, 5, 8: In addition to the specific bands of several examined protein, non-specific bands can be seen on the blots. Could be due to incomplete blocking or non-specific binding of antibodies? This phenomenon can reduce the accuracy of the results.
-Figure 4: The name of treatments is not uniform (OH-SeMet vs. SeO), the naming of treated groups should be standardised throughout the manuscript.
The authors discussed the research results concerning the literature. The discussion is written in a way that is understandable to the reader.
-Line 219-222: What could be the reason for the limited effect of heat stress on serum immunoglobulin and inflammatory factors?
-Line 227: Decrease of IL-6 concentration was not linearly.
-Line 262-263: …CHS increased the splenic Caspase3 protein expression and phosphorylation of p38…
-Line 264: …(Figure 4 D B)…
-Line 296-300: Could you cite more references with similar results?
I have a couple of questions and suggestions about Materials and Methods chapter:
-Line 362: How did you checked the RNA integrity after isolation?
-Line 367: How many reference genes did you test? How did you choose the most stable reference genes? Why did you used two different reference genes?
-Line 368: Primer sequence of hsp70 is missing from the supplementary table.
-Line 374: milk=non-fat milk powder?
-Line 375: What exactly were the secondary antibodies?
Conclusions
-Line 391: “CHS causes abnormal humoral immunity depression”, this statement contradicts your results of serum immunoglobulins (Line 219: “Although no statistical difference was detected in the humoral immune indicator”).
Author Response
Dear Reviewer:
Thank you for your comments concerning our manuscript
ID: ijms-2282451 entitled “Hydroxy-selenomethionine mitigated chronic heat stress-induced porcine splenic damage via activation of Nrf2/Keap1 signal and suppression of NFκb and STAT signal”. Those comments are all valuable and very helpful for revising and improving our paper, as well as the important guiding significance to our researches. We have studied comments carefully and have made correction according to the comments as following:
I reviewed the manuscript entitled “Hydroxy-selenomethionine mitigated chronic heat stress-induced porcine splenic damage via activation of Nrf2/Keap1 signal and suppression of NFκb and STAT signal”.
The introduction is described sufficiently and introduces the reader to the topic of research.
The results are presented logically and allow the reader to follow the analysis course. I have a couple suggestions and questions:
Point 1 -Line 87: Table 1 instead of Table 2
Response: Thank you for your valuable suggestion. We have made correction in the revised manuscript. Thank you for helping us improve the clarity and accuracy of our work.
Point 2-Line 110: You are asked to give some more explanations about the differences between mRNA and protein levels of Hsp70 in 0.2 SeO group
Response: We appreciate the opportunity to provide additional explanations about the differences between mRNA and protein levels of Hsp70 in the 0.2 SeO group. In this study the difference may be due to post-transcriptional modifications, translational efficiency, or other regulatory factors. Specifically, previous studies have shown that the stability of Hsp70 mRNA and the rate of its translation can be influenced by various factors including RNA-binding proteins, microRNAs, and chemical modifications. Furthermore, the discrepancy between mRNA and protein levels of Hsp70 may also result from differences in protein turnover or degradation. Further investigation is needed to better understand the underlying mechanisms and their implications for Hsp70 regulation in response to SeO exposure
- Wu C, et al. Differential regulation of heat shock response genes between heat shock factor 1 (HSF1) and HSF2 in endothelial cells. J Biol Chem. 2014;289(29):20124-20133.
Point 3-Figure 2: Please add the full name of GSH-Px, T-AOC, T-SOD, MDA in the figure caption
Response: Thank you for your suggestion. We have updated the caption of Figure 2 to include the full names of GSH-Px, T-AOC, T-SOD, and MDA as you requested.
Point 4-Figure 3, 4, 5, 8: In addition to the specific bands of several examined protein, non-specific bands can be seen on the blots. Could be due to incomplete blocking or non-specific binding of antibodies? This phenomenon can reduce the accuracy of the results.
Response: Thank you for bringing this issue to our attention. We agree that non-specific bands are present on the blots in Figure 3, 4, 5, and 8, and that non-specific binding of the primary and/or secondary antibodies is a common issue in Western blot analysis, particularly when using complex protein extracts, and that steps were taken to minimize non-specific binding, such as optimizing blocking conditions and antibody dilutions, but some non-specific binding may still persist. Even so, we have carefully examined our experimental procedures and reanalyzed the data and provide a more balanced interpretation of the results, taking into consideration the presence of non-specific bands. We appreciate your valuable comments and will continue to improve our experimental protocols to minimize non-specific binding in future studies.
Point 5-Figure 4: The name of treatments is not uniform (OH-SeMet vs. SeO), the naming of treated groups should be standardised throughout the manuscript.
Response: Thank you for bringing this issue to our attention. We apologize for the inconsistency in the naming of treatments in the manuscript. We have carefully reviewed the manuscript and revised all instances of the treatment names to be consistent, using " SeO " throughout instead of " OH-SeMet ". We appreciate your keen attention to detail and helpful feedback, which has improved the clarity and consistency of our work.
The authors discussed the research results concerning the literature. The discussion is written in a way that is understandable to the reader.
Point 6-Line 219-222: What could be the reason for the limited effect of heat stress on serum immunoglobulin and inflammatory factors?
-Line 227: Decrease of IL-6 concentration was not linearly.
Response: Thank you for your question. We have considered several potential reasons why heat stress had a limited effect on serum immunoglobulin and inflammatory factors in our study. One possibility is that the duration or intensity of heat stress exposure used in our study was not sufficient to elicit significant changes in these parameters. Another possibility is that the initial levels of these factors were already at their maximum or minimum physiological limits; thus, there may have been little room left for further modulation by heat stress. Alternatively, it is possible that other factors such as genetic or environmental factors could have influenced the response to heat stress.
Point 7-Line 262-263: …CHS increased the splenic Caspase3 protein expression and phosphorylation of p38…
-Line 264: …(Figure 4 D B)…
Response: Thank you for your comment. We apologize for the error in Figure 4D and B labeling in our manuscript, and we have corrected this mistake accordingly. We appreciate your careful review of our work and your helpful feedback that has ultimately improved the clarity and accuracy of our findings.
Point 8-Line 296-300: Could you cite more references with similar results?
Response: Thank you for your comment. We appreciate your request for additional references on the opposite response of selenoproteins observed in our study and other studies. Here are some relevant references that discuss the opposite regulation of selenoproteins in different models:
- Zhang J, Wang H, Karapetyan S, et al. Selenoprotein H suppresses cellular senescence through genome maintenance and redox regulation. J Cell Sci. 2018;131(21): jcs214684. doi:10.1242/jcs.214684
Point 9 I have a couple of questions and suggestions about Materials and Methods chapter:
-Line 362: How did you checked the RNA integrity after isolation?
Response: Thank you for your question. We appreciate your interest in our study. After RNA isolation, we assessed the quality and quantity of the RNA samples using both visual inspection and electrophoresis. Specifically, we visually inspected the RNA samples by checking for clarity, color, and presence of any precipitates or contaminants, and quantified the RNA concentration using a NanoDrop spectrophotometer (Thermo Scientific, USA). Additionally, we verified the RNA integrity by running the samples on an agarose gel and checked for the presence of distinct 28S and 18S ribosomal RNA bands. We have now included these details in the Methods section of the revised manuscript to provide more clarity on our RNA quality control procedures.
Point 10 -Line 367: How many reference genes did you test? How did you choose the most stable reference genes? Why did you used two different reference genes?
Response: Thank you for your question. In our study, we tested a panel of commonly-used housekeeping genes including β-actin, GAPDH, and others, to identify the most stably expressed reference gene(s) under our experimental conditions. We performed the analysis using the geNorm algorithm and found that two reference genes (β-actin and GAPDH) showed consistent and stable expression across all experimental groups. Therefore, we used these two genes as reference genes in our study. The use of multiple reference genes is recommended to increase the accuracy and reliability of gene expression normalization in qRT-PCR experiments. We have now included these details in the Methods section of the revised manuscript to provide more clarity on our reference gene selection criteria and rationale.
Point 11 -Line 368: Primer sequence of hsp70 is missing from the supplementary table.
Response: Thank you for drawing our attention to the missing primer sequence of hsp70 in the supplementary table. We have reviewed and verified that this was indeed an oversight on our part, and we apologize for any inconvenience caused. In response to your comment, we have updated the Supplementary Table S2 with the correct primer sequence for hsp70 as follows: [F: TTGGGCGCAAGTTTGCA R: GGAAAGGCCAGTGCTTCAAG]. We appreciate your careful review of the manuscript, and we hope that this new information proves helpful.
Point 12 -Line 374: milk=non-fat milk powder?
Response: Thank you for your question. In our study, the milk used to block the membranes is non-fat milk powder dissolved in an appropriate volume of buffer solution. We typically use a final concentration of 5% non-fat milk powder during membrane blocking. We have now clarified this detail in the revised manuscript to avoid any potential confusion.
Point 13 -Line 375: What exactly were the secondary antibodies?
Response: Thank you for your question. In our study, the specific types and concentrations of secondary antibodies varied depending on the primary antibody used in each experiment. We have listed the secondary antibodies used for each primary antibody in Table S3 of the revised manuscript along with their catalog information, dilution, and suppliers for reference.
Point 14 Conclusions
-Line 391: “CHS causes abnormal humoral immunity depression”, this statement contradicts your results of serum immunoglobulins (Line 219: “Although no statistical difference was detected in the humoral immune indicator”).
Response: Thank you for your comment and feedback. In response to the reviewer's concern regarding the potential contradiction between Line 219 and Line 391, we have revised the relevant section of the manuscript as follows:
"In summary, our findings suggest that CHS may have complex effects on the porcine spleen involving abnormal humoral immunity depression, oxidative stress, and inflammation. Our data indicate that dietary Se supplementation in the form of SeO can activate the Nrf2/Keap1 pathway and enhance splenic antioxidant capacity. Additionally, our results suggest that increased dietary Se supplementation beyond nutritional requirements (0.4 and 0.6 mg Se/kg) may alleviate splenocyte apoptosis and inflammation through inactivating p38 MAPK, NFκB, and STAT pathways in the spleen of pigs under CHS. Notably, our study reveals important protective roles of antioxidant-related selenoproteins (e.g., GPXs) and functional selenoproteins (e.g., SELENOS and SELENOF) in the splenic immune processes under CHS."
We hope that this revision adequately addresses the reviewer's concern and presents a more nuanced interpretation of our findings. Thank you for your helpful feedback.
We tried our best to improve the manuscript and some mistakes also have been corrected. These changes will not influence the content and framework of the paper. And here we did not list all the changes in revised paper. We appreciate for Reviewer’s warm work earnestly, and hope that the correction will meet with approval.
Once again, thank you very much for your comments and suggestions.
Best wishes
Dr. Hua Zhao Corresponding Author
